# Modular access to alkylgermanes via reductive germylative alkylation of activated olefins under nickel catalysis

Rui Gu[1,2], Xiujuan Feng[2], Ming Bao ®[2] ✉ & Xuan Zhang ®[1,2] ✉

Carbon-introducing difunctionalization of C-C double bonds enabled by transition-metal catalysis is one of most straightforward and efficient strategies to construct C-C and C-X bonds concurrently from readily available feedstocks towards structurally diverse molecules in one step; however, analogous difunctionalization for introducing germanium group and other functionalities remains elusive. Herein, we describe a nickel-catalyzed germylative alkylation of activated olefins with easily accessible primary, secondary and tertiary alkyl bromides and chlorogermanes as the electrophiles to form C-Ge and C-C$_{alkyl}$ bonds simultaneously. This method provides a modular and facile approach for the synthesis of a broad range of alkylgermanes with good functional group compatibility, and can be further applied to the late-stage modification of natural products and pharmaceuticals, as well as ligation of drug fragments. More importantly, this platform enables the expedient synthesis of germanium substituted ospemifene-Ge-OH, which shows improved properties compared to ospemifene in the treatment of breast cancer cells, demonstrating high potential of our protocol in drug development.

Due to their abundance and widespread availability of alkenes, difunctionalizaiton processes that introduce two functional groups on each side of C-C double bonds[1–12] has emerged as a versatile and powerful tool for rapid construction of high-value and complex molecular scaffolds in a single operation. Over the past decade, transition-metal catalysis[13–20] has drawn great attention, because it enables these reactions to proceed with good selectivity and high efficiency. Among them, carbonized difunctionalization of olefins[21–26] is well developed (Fig. 1a, path a), which provides attractive manipulations for creation of C-C and C-X bonds simultaneously. Despite such impressive advances, employing carbon group elements in difunctionalizaiton of alkenes under transition-metal catalysis, such as germanium[27–29] for preparing organogermanes (Fig. 1a, path b), remains underdeveloped.

Germanium is in the same column of periodic table as carbon atom, but possesses different electronegativity and atom radius. Therefore, Germanium is generally considered as a bioisostere of carbon in medical chemistry[30,31] for biological and pharmacological studies, because of their relatively high robustness, hydrophobicity and low toxicity. In addition, organogermanes have also been applied to functional material areas[32–34], such as molecular electronics, nanoscale lithography, and biosensors. Aside from above applications, organogermanium compounds have led to increased interest in the investigation of their reactivities in organic synthesis and catalysis[35–42]. As such, synthetic access to germanium-containing architectures by a general and effective way is highly appealing, but challenging. Traditionally, construction of these molecules predominantly relies on organometallic reactivity through the nucleophilic substitution of a

[1]School of Chemistry and Materials Science, Institute of Advanced Materials and Flexible Electronics (IAMFE), Nanjing University of Information Science and Technology, 219 Ningliu Road, Nanjing 210044, China. [2]State Key Laboratory of Fine Chemicals, School of Chemical Engineering, Dalian University of Technology, 2 Linggong Road, Dalian 116024, China. ✉e-mail: mingbao@dlut.edu.cn; xuanzhang@nuist.edu.cn

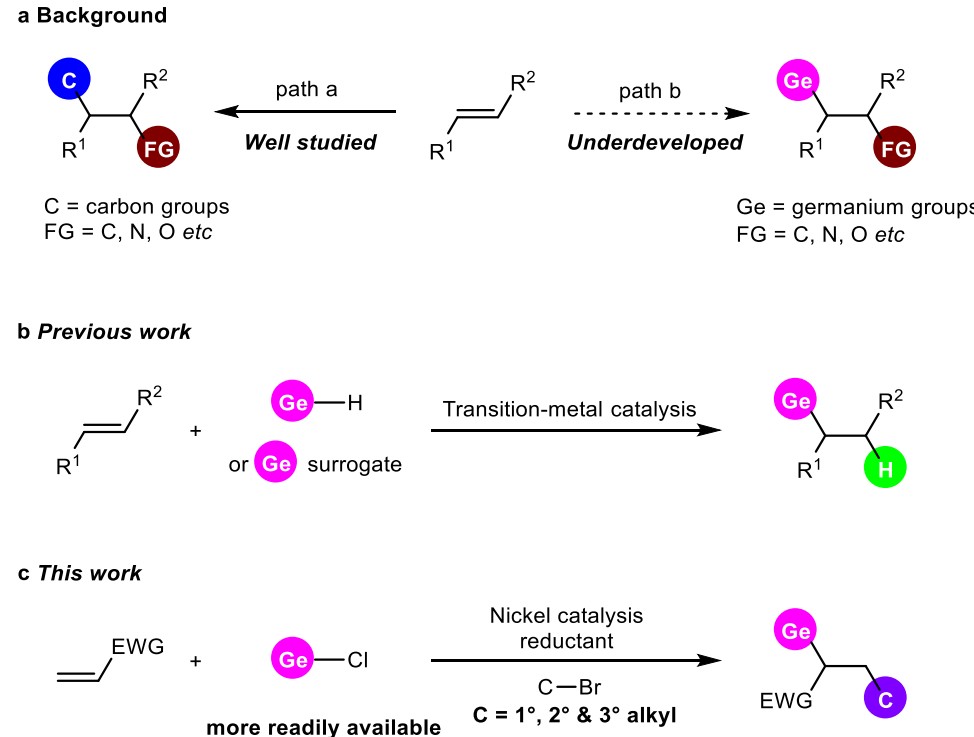

**a Background**

C = carbon groups
FG = C, N, O *etc*

Ge = germanium groups
FG = C, N, O *etc*

**b Previous work**

**c This work**

**Fig. 1 | Difunctionalization of alkenes and germylative functionalization of alkenes. a** difunctionalization of alkenes. **b** germylative hydrogenation of alkenes. **c** reductive germylative alkylation of alkenes. EWG electron-withdrawing groups. The carbon groups are colored in blue. The common functional groups are colored in brown. The germanium groups are colored in pink. The hydrogen atom is colored in green. The alkyl groups are colored in purple.

germyl electrophile by Grignard[43] or organolithium[44] reagents or, vice versa, by a nucleophilic germanium reagent[45–47] reacting with electrophiles. However, the need of pre-generation and difficulty in handling of organometallic reagents, and their high reactivity without discrimination undoubtedly limit the functional group compatibility. Catalytic approaches for synthesizing organogermanes including cross-coupling reactions[48–52], C-H bond germylations[53–56] have not been achieved until recently, where palladium, rhodium or nickel catalysis were employed. These processes typically use Me$_3$Ge-GeMe$_3$, R$_3$GeH, or R$_3$Ge-Zn as the germyl reagents. Despite such formidable progresses, their synthetic applications are still restricted by several critical disadvantages such as multiple-step synthesis, limited substrate scope and hardly available germyl sources. Hence, given the importance of expanding the chemical space of alkene difunctionalization and providing a synthetic toolbox to create germanium-substituted molecules with diverse and complex structures for exploring their application potential, developing a new strategy through germylative transformation of olefins to install germyl groups into organic motifs with easily accessible substrates and convenient germyl sources would be greatly attractive and beneficial.

Works to construct C-Ge bonds via alkene difunctionalization have emerged over past years[27–29,57–65]. But it's worth noting that transition-metal catalyzed hydrogermylations are most frequently studied[57–62], which introduce a germyl group on one side of C-C double bond and only a hydrogen atom on the other position (Fig. 1b). These monofunctionalized protocols largely reduce the complexity and diversity of germanium-containing molecules, and consequently diminish their further application. In contrast, germylative alkylation of alkenes enabled by transition-metal catalysis would be much more fascinating but still unexplored. The introduction of alkyl groups can not only significantly increase structural complexity of molecules, but also work as useful descriptors for greater three-dimensional shape compared to flat arenes, and more likely to succeed as drug candidates because of their better match with biological targets which have 3D

structures[66–68]. Herein, we report a nickel catalyzed reductive germylative alkylation of activated olefins using both readily available unactivated alkyl bromides and chlorogermanes as the starting materials (Fig. 1c). In this work, a modular and general method under mild conditions to access alkylgermanes with high molecular diversity is presented, which exhibits broad substrate scope, good functionality tolerance and high potential for improving efficacy of drug via structure modification.

## Results and discussion

### Reaction discovery

To evaluate the desired reaction, we selected (3-bromopropyl)benzene **1a** as the model alkyl source, ethyl acrylate **2a** as the alkene unit, chlorotrimethylgermane **3a** as the germanium reagent, NiBr$_2$•DME as the precatalyst, 4,4'-di-*tert*-butyl-2,2'-bipyridine **L1** as the ligand and manganese powder as the reductant to initiate the optimization studies. The mixture was stirred in *N*,*N*-dimethyl acetamide (DMA) at 35 °C for 36 h under a N$_2$ atmosphere to afford the desired product ethyl 6-phenyl-2-(trimethylgermyl)hexanoate **4** in 15% assay yield, along with 41% of recovered **1a** (Table 1, entry 1) and side products including hydro-debromination of **1a**, dehydrobromination of **1a** and hydroalkylation of **2a** (for details, see the Supplementary Table 1). Encouraged by the promising result, we further investigated a variety of ligands with different skeletons; compared with pyridine-containing ligands, the bisoxazoline ligand **L5** gave the best reactivity with good mass balance (Table 1, entries 2–5). Then, a set of nickel precatalysts were examined in the presence of **L5**, and NiBr$_2$ was found to be optimal, which improves the yield of product **4** to 73% (Table 1, entries 6–9). Surprisingly, the reaction proceeds better in the absence of ligand **L5** using NiBr$_2$ as the catalyst to give a yield up to 86% yield (Table 1, entry 10). Subsequently, solvent effects were studied under the ligand-free conditions, and the observations indicate the solvent has a big impact on the reaction performance. Among the solvents studied, DMA represents the best choice, and other solvents, such as

**Table 1 | Optimization of reaction conditionsᵃ**

| Entry | Cat. Ni | Ligand | Reductant | Solvent | Yield of 4 (%) | Recovery of 1a (%) |
|---|---|---|---|---|---|---|
| 1 | NiBr₂•DME | L1 | Mn | DMA | 15 | 41 |
| 2 | NiBr₂•DME | L2 | Mn | DMA | 4 | 1 |
| 3 | NiBr₂•DME | L3 | Mn | DMA | 19 | 57 |
| 4 | NiBr₂•DME | L4 | Mn | DMA | 3 | 62 |
| 5 | NiBr₂•DME | L5 | Mn | DMA | 35 | 50 |
| 6 | NiBr₂ | L5 | Mn | DMA | 73 | 7 |
| 7 | Ni(acac)₂ | L5 | Mn | DMA | 30 | 50 |
| 8 | Ni(PPh₃)₂Cl₂ | L5 | Mn | DMA | 5 | 87 |
| 9 | Ni(COD)₂ | L5 | Mn | DMA | 37 | 49 |
| 10 | NiBr₂ | — | Mn | DMA | 86 (80ᵇ) | 2 |
| 11 | NiBr₂ | — | Mn | DMSO | trace | 37 |
| 12 | NiBr₂ | — | Mn | CH₃CN | 4 | 91 |
| 13 | NiBr₂ | — | Mn | EtOAc | 0 | 100 |
| 14 | NiBr₂ | — | Zn | DMA | 6 | 88 |
| 15 | NiBr₂ | — | TDAE | DMA | 1 | 43 |
| 16ᶜ | NiBr₂ | — | Mn | DMA | 53 | 25 |
| 17ᵈ | NiBr₂ | — | Mn | DMA | 71 | 1 |
| 18ᵉ | NiBr₂ | — | Mn | DMA | 77 | 1 |

ᵃReaction conditions: **1a** (0.1 mmol), **2a** (0.15 mmol), **3a** (0.15 mmol), Ni catalyst (10 mol%), ligand (12 mol%), reductant (0.3 mmol), solvent (1 mL), 35 °C, 36 h, N₂ atmosphere. Yields are determined by GC using dodecane as the internal standard.
ᵇThe isolated yield was shown in parentheses on 0.3 mmol scale.
ᶜNiBr₂ (5 mol%) was used.
ᵈ**2a** (0.12 mmol) was used.
ᵉ**3a** (0.12 mmol) was used. DME dimethyl ether, DMA N,N-dimethylacetamide, DME dimethyl ether, DMA N,N-dimethylacetamide, acac acetylacetonate, COD 1,5-cyclooctadiene, DMSO dimethyl sulfoxide, EtOAc ethyl acetate, TDAE tetrakis(dimethylamino)ethylene.

dimethyl sulfoxide (DMSO), acetonitrile, ethyl acetate (EtOAc) almost suppress the reaction (Table 1, entries 11-13). After that, several reductants were screened; zinc powder and tetrakis(dimethylamino) ethylene (TDAE) show much lower reactivity, and manganese remains the suitable reagent (Table 1, entries 14 and 15). Reducing the catalytic loading of NiBr₂ resulted in decreased yield of 53% (Table 1, entry 16). The utilization of less amount of alkene **2a** or chlorogermane **3a** also lowered the reaction efficiency (Table 1, entries 17 and 18). Ultimately, the optimized conditions were identified as follows: NiBr$_2$ (10 mol%) and Mn (3 equiv.) in DMA (0.1 M) at 35 °C for 36 h (Table 1, entry 10).

## Substrate scope

With the optimized set of reaction conditions in hand, the scope of this new transformation was then investigated (Fig. 2). Choosing ethyl acrylate **2a** and chlorotrimethylgermane **3a** as the representative substrates, the generality of unactivated alkyl halides was explored first. Besides (3-bromopropyl)benzene **1a**, (3-iodopropyl)benzene **1a'** and (3-chloropropyl)benzene **1a''** were also investigated under standard conditions, in which different reactivities were obtained for synthesis of product **4** (52% yield from **1a'** and no product from **1a''**).

Various (2-bromoethyl)benzene derivatives **1b-1g** including electron-donating groups (-OMe and -NHBoc) and electron-withdrawing groups (-Cl, -Br, -CO$_2$Me and -CF$_3$) substituted on the phenyl ring were compatible, which smoothly underwent germylalkylation reactions to produce desired products **5-10** in good yields (58%-80%). Given the prevalence of heteroaryl rings in drug molecules, we were delight to find that a number of heterocycle-containing alkyl bromides could also be successfully coupled in this reaction. A range of five- and six-membered heteroarenes, such as thiophene, benzofuran, indole, carbazole and quinoline all survived well in the reductive conditions (**11-15**). Notably, substrate with coordinating Lewis basic N($sp^2$) atom (**15**) can work with slightly modified conditions. All-carbon alkyl bromides, 1-bromobutane **1m** and (bromomethyl)cyclobutene **1n** are undoubtedly suitable substrates under the three-component reaction conditions to afford corresponding alkylgermanes **16** and **17** in good yields (64% and 72%). The conditions were also applicable to a plenty of valuable functionalized units (**18-31**) including some potentially sensitive functional groups and useful synthetic handles, such as ketone (**22**), acetal (**27**), tertiary amine (**30**), chloride (**20**), nitrile (**23**), alkenes (**25** and **26**), phosphate (**28**), sulfone (**29**) and boronic ester (**31**), which

**Fig. 2 | The scope of alkyl bromides[a].** [a]Reaction conditions: **1** (0.3 mmol), **2a** (0.45 mmol), **3a** (0.45 mmol), NiBr₂ (10 mol%), Mn (0.9 mmol), DMA (3 mL), 35 °C, 36 h, N₂ atmosphere. Isolated yield was shown. [b](3-iodopropyl)benzene was used instead of (3-bromopropyl)benzene. [c](3-chloropropyl)benzene was used instead of (3-bromopropyl)benzene. [d]L5 (12 mol%) was used as the additive. [e]diastereomeric ratio can't be determined by Proton Nuclear Magnetic Resonance and High Performance Liquid Chromatography. Boc tert-butyloxycarbonyl, Ts tosyl.

can provide a good chance to produce more structurally diverse organogermanes after corresponding modifications. Notably, reactive functional groups such as boronic esters, which is commonly problematic in transition-metal-catalyzed reactions, could also be tolerated well in this reaction (**31**). Then, we turn to explore secondary and tertiary alkyl bromides. The reaction of 3-bromopentane **1af** with alkene **2a** and chlorogermane **3a** proceeded smoothly under standard conditions to deliver desired germylated alkyl carboxylic ester **32** in 76% yield. Cycloalkyl groups with ring sizes ranging from four to six and 2-adamantyl were all amenable in our system to give carbocycle derivatives **33-36** in good yields (76%-95%). Moreover, alkyl bromides **1ak-1am** bearing various saturated heterocycles were studied, which were demonstrated to be suitable substrates with good efficiency (**37-39**, 64%-80%). Meanwhile, acyclic and cyclic tertiary carbons on alkyl bromides were successfully tolerated in this protocol to construct quaternary carbon center-containing alkyl germanium compounds **40–45** in decent yields (63%-78%). Groups such as hydroxyls, aldehydes, alkynes and amides were not amenable in current system, which gave the target products in <10% yields (for details, see the Supplementary Information).

Next, we focused on evaluating the scope of activated olefins and chlorogermanes utilizing (3-bromopropyl)benzene **1a** as the alkylating reagent (Fig. 3). A set of acrylates **2b-2d** underwent the target transformation smoothly, providing the products **46-48** in 61%-76% yields. Other types of activated alkenes like acrylamide **2e**, acrylonitrile **2f**, (vinylsulfonyl)benzene **2g** and diethyl vinylphosphonate **2h** were also amenable to the reaction with reasonable efficiency under slightly modified conditions (**49-52**, 34%-68%). In addition, steric hindered methyl methacrylate **2i** and dimethyl itaconate **2j** could be transformed into quaternary carbon centers substituted with a germyl group in synthetically useful yields (**53** and **54**). Unfortunately, β-substituted acrylates including methyl (*E*)-but-2-enoate, 5,6-dihydro-2H-pyran-2-one and methyl cyclopent-1-ene-1-carboxylate as well as styrene were all incompatible substrates (for details, see the Supplementary Information). Followingly, the reactivity of chlorogermanes were examined by varying substituents on germanium atom. As expected, the reactions of chlorotrialkylgermanes **3b-3d** possessing longer alkyl chain can proceed with similar efficiency compared to the standard reaction employing chlorotrimethylgermane **3a** (**55-57**). The arylated germanium-containing products **59-60** were observed in good to high yields with phenyl substituted chlorogermanes **3e-3g** as the starting materials under optimal conditions. These results indicate there is no obvious substituent effect on germanium reagent.

Taking account of the broad substrate scope and good functional group compatibility of this protocol, we were encouraged to use the developed reaction for modifying drug molecules (Fig. 4). Alkyl bromides derived from commercial drugs, such as ospemifene, probenecid, (1*R*)-(+)-Camphanic acid, oxaprozin and triclosan were all straightforwardly modified through the three-component reaction with alkene **2a** and chlorogermane **3a**, which opens a new direction for downstream derivatizations (**61-65**). On the other hand, pharmaceutically relevant olefins were also subjected to the reaction conditions. For example, olefins bearing L-menthol (**66**) and indomethacin (**67**) were coupled in 66% and 46% yields, respectively. Lastly, complex molecule **68** having two pharmacophores (L-menthol and probenecid) was easily obtained with this transformation, demonstrating the applicable potential of our protocol in drug discovery.

## Synthetic applications

The practicality of the germylative alkylation reaction was showcased by gram-scale reaction of model substrates, delivering target product **4** in 1.33 g with similar efficiency (79% versus 80%). The β-hydroxy alkylgermane **69** was directly synthesized from the reduction of **4** in the presence of LiAlH$_4$ (Fig. 5a). Moreover, the alkylgermane **4** can be converted into β-hydroxy carboxylate **70** in the presence of TBAF via degermylative addition with benzaldehyde (Fig. 5a), which is a valuable building blocks. Ethyl 2,6-dibromo-6-phenylhexanoate **71**, bearing similar structure with a classical synthon-α-bromo carboxylate, was accessed through the degermylative dibromination of **4** (Fig. 5a). Ospemifene (Osphena), approved by the U.S. Food and Drug Administration, is used for the treatment of moderate to severe dyspareunia (vaginal and vulvar atrophy associated with menopause). It has shown the bioactivity in the prevention of breast cancer[69,70] as a novel Selective Estrogen Receptor Modulator (SERM) that acts as an agonist in brain, bone and vagina and acts as an antagonist in uterus and breasts, which possesses ideal characteristics of a SERM. Although this drug has exhibited a promising pharmacological profile, the high value of half-maximal inhibitory concentration (IC$_{50}$) limits its application and encourages us to modify the structure of ospemifene to pursue better efficacy. To demonstrate the potential of prepared alkylgermanes and the applicability of our method to drug discovery settings, we subsequently applied our new germylative alkylation reaction to the synthesis of germanium-containing ospemifene analog (Fig. 5b). Toward this end, ospemifene-Ge-OH **72** was afforded by **1bq** coupling with **2a** and **3a**, and followed by reduction (two steps, 35% overall yield). The synthesized ospemifene analog **72** was evaluated for its

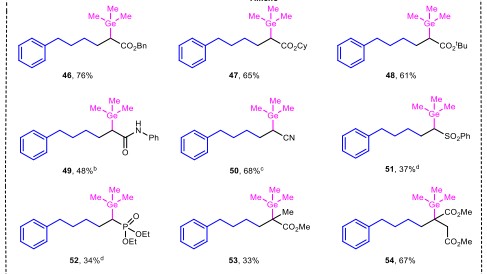

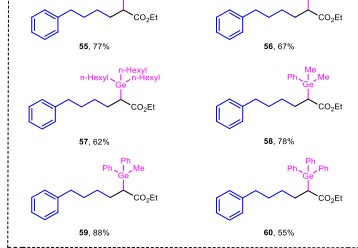

**Fig. 3 | The scope of alkenes and germanium reagents**$^a$. $^a$Reaction conditions: **1a** (0.3 mmol), **2** (0.45 mmol), **3** (0.45 mmol), NiBr$_2$ (10 mol%), Mn (0.9 mmol), DMA (3 mL), 35 °C, 36 h, N$_2$ atmosphere. Isolated yield was shown. $^b$**L5** (12 mol%) was used as the additive. $^c$NiBr$_2$ (15 mol%), **L5** (20 mol%) was used at 50 °C, 60 h. $^d$NiBr$_2$ (15 mol%) was used. EWG electron-withdrawing group, Bn benzyl, Cy cyclohexyl, $^t$Bu *tert*-butyl.

**Fig. 4 | The late-stage functioanlization[a].** [a]Reaction conditions: **1** (0.2 mmol), **2** (0.3 mmol), **3a** (0.3 mmol), NiBr$_2$ (10 mol%), Mn (0.6 mmol), DMA (2 mL), 35 °C, 36 h, N$_2$ atmosphere. Isolated yield was shown. EWG electron-withdrawing group, *i*-Pr *iso*-propyl.

anti-proliferative activities against MCF-7 (ER+) and MDA-MB-231 (ER-) human breast cancer cell-lines using MTT assay (Fig. 5c, d). The IC$_{50}$ values showed that compound **72** was more effective against MCF-7 and MDA-MB-231 cells compared to ospemifene (45.6 μM versus 77.2 μM in MCF-7 cells, 59.58 μM versus 115.96 μM in MDA-MB-231 cells). As control, the IC$_{50}$ value of degermylative ospemifene-OH **73** were also measured, which were 94.09 μM for MCF-7 and 98.70 μM for MDA-MB-231 cells independently. All collected results clearly indicated that the improved anti-cancer activity is resulted from the installation of germanium group. In addition, the metabolic stability of **72** and ospemifene were explored by mouse liver microsomal metabolic assay (Fig. 5e). After 45 min of incubation in mouse microsomes, the amount of remaining **72** was higher than that of ospemifene. The calculated half-life of **72** (t$_{1/2}$ = 13.90 min) was ~3.6 times than that of ospemifene (t$_{1/2}$ = 3.82 min), suggesting the metabolic stability of ospemifene in mouse microsomes was enhanced after modification. To evaluate and compare the toxicity against normal cell between ospemifene and compound **72**, MCF-10A was chosen as the normal human cell line for studying. As shown in Fig. 5f, the semi-lethal doses of ospemifene and compound **72** against MCF-10A cell line were 53.25 μM and 158.36 μM respectively, which implied the germanium-modified molecule **72** has less toxic side effects. Lastly, we carried out a hemolysis experiment against red blood cells to access the hemocompatibility (see the Supplementary Notes and Fig. S1). Negligible hemolytic activities were observed at 120 mM for ospemifene and **72** (both of hemolysis percentage are <8%), indicating the installation of germanium group has no impact on the blood compatibility. Altogether, these results demonstrate the high potential of this germanium-introducing strategy for improving the pharmacological properties of drug candidates.

## Mechanism studies

To shed light on the possible mechanism of this reductive germylative alkylation reaction, a set of control experiments were conducted (Fig. 6). No desired product **4** was observed under model reaction conditions by adding stoichiometric amount of a radical scavenger, 2,2,6,6-tetramethyl-1-piperidinoxyl (TEMPO). Instead, radical trapping product **74** was detected by high-resolution mass spectrometry (Fig. 6a). Next, two radical-clock experiments were conducted with 6-

bromohex-1-ene (**1bv**) and (bromomethyl)cyclopropane (**1bw**) as the alkyl reagents, which are well-known radical probes. Cyclization product **76** and ring-opening products **77, 78** were isolated in 42% and combined 62% yields, respectively. Both results indicate that an alkyl radical is produced from the alkyl bromide and involved in the reaction as a key species (Fig. 6b). In the presence of 1 equiv. Ni(COD)$_2$, the reaction of **1a, 2a** and **3a** in DMA at 35 °C for 36 h led to no product **4**, along with 38% of **1a** consumed. In another experiment, the catalytic reaction using 10 mol% Ni(COD)$_2$ proceeded successfully to afford target product **4** in 31% yield (Fig. 6c). these observations suggest: (1) alkyl bromide should react with nickel catalyst firstly in the reaction; (2) the manganese powder not only works as a terminal reductant, but also participates in a reductive process to generate a key reactive intermediate in the catalytic cycle.

Based on the preliminary results and previous reports[51,52], a radical-type mechanism is tentatively proposed in Fig. 6d. The reaction of alkyl bromide **1** with Ni (0) generated in-situ by reduction of manganese, affords Ni (I) and an alkyl radical, which is then captured by activated olefin **2** to produce a new carbon-based radical **A**. The species **A** reacts with Ni (I) to give an alkyl-Ni (II) intermediate **B**, followed by reduction by manganese for giving a reactive alkyl-Ni (I) **C**. The oxidative addition of chlorogermane **3** with species **C** may affords alkyl-Ni (III) complex **D**, which undergoes reductive elimination to deliver the desired product.

In conclusion, we have developed a catalytic reductive germylative alkylation of activated olefins enabled by nickel catalysis under mild conditions, delivering alkylgermanes with high efficiency. This method features the direct utilizing of easily accessible alkyl bromides and chlorogermanes as the reaction partners, avoiding pre-preparation of sensitive organometallics that are often tedious to handle. The developed approach can tolerate a broad range of functional groups as well as natural products and pharmaceuticals, thereby provides a powerful tool towards highly structure-diverse aliphatic germanes including previously inaccessible motifs. Furthermore, the better pharmacological properties of the germanium-modified ospemifene compound **72** against breast cancer cells were observed by a number of biological experiments, highlighting the synthetic value and power of the present protocol. Given the generality and practicality of

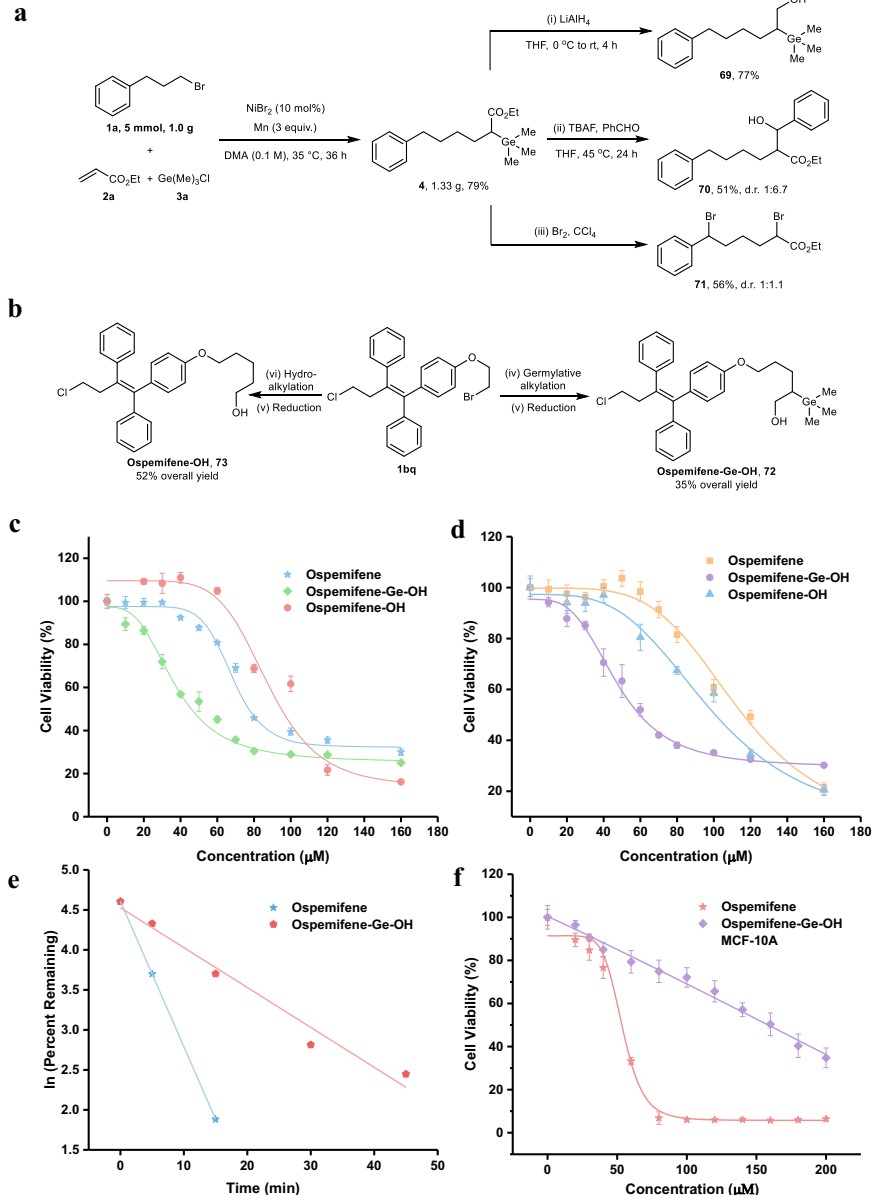

**Fig. 5 | Synthetic applications. a** gram-scale reaction of **1a**, **2a** and **3a**, and transformations of **4**; **4** (0.2 mmol), LiAlH₄ (0.1 mmol), THF (2 mL), 0 °C, 4 h, N₂ atmosphere; (ii) **4** (0.2 mmol), TBAF (0.24 mmol), PhCHO (0.24 mmol), THF (0.4 mL), 45 °C, 24 h, N₂ atmosphere; (iii) **4** (0.2 mmol), Br₂ (0.2 mmol), CCl₄ (10 mL), 0 °C to room temperature, 3.5 h, N₂ atmosphere; (**b**) synthetic route of compounds **72** and **73**; (iv) standard condition; (v) **61** (0.2 mmol), LiAlH₄ (0.1 mmol), THF (2 mL), 0 °C, 4 h, N₂ atmosphere; (vi) **1bq** (1 mmol), NiCl₂(PPh₃)₂ (0.1 mmol), Zn (2.5 mmol), ethyl acrylate (4 mmol), H₂O (1 mmol), CH₃CN (2.5 mL), 80 °C, 12 h, N₂ atmosphere; (vii)

Hydroalkylation product (0.2 mmol), LiAlH₄ (0.1 mmol), THF (2 mL), 0 °C, 4 h, N₂ atmosphere. **c** the MTT assays of **72, 73** and ospemifene against MCF-7 cells; (**d**) the MTT assays of **72, 73** and ospemifene against MDA-MB-231 cells; (**e**) mouse liver microsomal metabolic assays of **72** and ospemifene; **f** the MTT assays of **72** and ospemifene against MDA-10A. Experiments details were shown in the Supplementary Notes 3.3 Synthetic Applications. Data are presented as mean values ± SD (*n* = 3) biologically independent samples for panels (**c, d, f**).

this alkene difunctionalization strategy, it opens a new window for organogermanes synthesis and is expected to stimulate substantial applications to functional materials and drug discovery.

## Methods

### General procedure for Ni-catalyzed germylative alkylation of activated olefins

The procedure was conducted in a nitrogen-filled glove box. In an oven-dried 8 mL reaction vial equipped with a magnetic stir bar, alkene (0.45 mmol, 1.5 equiv.), germanium reagent (0.45 mmol, 1.5 equiv.), NiBr₂ (10 mol%), Mn (0.9 mmol, 3.0 equiv.) and DMA (3 mL, 0.1 M) were charged and pre-stirred at 35 °C for 0.5 h, then alkyl

bromine (0.3 mmol, 1.0 equiv.) was added under N₂. The reaction vial was sealed and removed from the glove box. the mixture was stirred at 35 °C for another 36 h, subsequently quenched with water (10.0 mL) and extracted with dichloromethane (3 × 15.0 mL). The combined organic layers were washed with water, brine, dried over anhydrous Na₂SO₄, and concentrated under reduced pressure. The residue was purified by flash chromatography on silica gel to afford product.

### Reporting summary

Further information on research design is available in the Nature Portfolio Reporting Summary linked to this article.

**Fig. 6 | Mechanism studies. a** radical trapping experiment. **b** radical-clock experiments. **c** control experiments. **d** proposed mechanism. Reaction details were shown in the supplementary notes 3.4 Mechanistic Studies. TEMPO 2,2,6,6-tetramethyl-1-piperidinoxyl, COD 1,5-cyclooctadiene, EWG electron-withdrawing group.

## Data availability

The authors declare that the data supporting the findings of this study, including experimental details and compound characterization, are available within the article and its Supplementary Information file or source data file. Source data are provided with this paper. All other data are available from the corresponding author upon request.

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

## Acknowledgements

We gratefully acknowledge the financial support of the "Thousand Talents Plan" Youth Program (X.Z.), the "Jiangsu Specially-Appointed Professor Plan" (R2020T30, X.Z.), the "Innovation & Entrepreneurship Talents Plan" (X.Z.), the National Natural Science Foundations of China (22101136, X.Z. and 22172014, M.B.), LiaoNing Revitalization Talents Program (XLYC1802030, M.B.) and the Natural Science Foundation of Jiangsu Province (BK20200806, X.Z.). We also thank Prof. Guangzhe Li (Dalian University of Technology) for supplying breast cancer cells and experimental instruments, Shanghai Medicilon company for the mouse liver microsomal metabolic assay and Prof. Andrew McNally (Colorado State University) for assistance with the manuscript proofing.

## Author contributions

X.Z. conceived the project. R.G. conducted experiments. X.Z. and M.B. supervised the research and wrote the manuscript with the assistance of other authors. All the authors analyzed the data.

## Competing interests

The authors declare no competing interests.
