## [Peer Review File · Nature Communications]

Modular Access to Alkylgermanes via Reductive Germylative Alkylation of Activated Olefins Under Nickel CatalysisReviewer #1:

Remarks to the Author:

In this manuscript, Gu and co-workers reported a reductive gemylative alkylation of activated olefins with unactivated alkylbromides and chlorogermanes enabled by nickel catalysis for construction of various alkylgermanes. Compared to well-known hydrogermylation reactions, this method can afford alkylgermanes with much higher structural complexity and diversity. On the other hand, chlorogermanes were used as the germanium sources instead of generally employed germymetallic reagents, which can avoid preparing and handling of air and moisture sensitive organometallic reagents and further show much better functional group tolerance. Significantly broad substrate scope was demonstrated with respect to alkyl bromides, alkenes and chlorogermanes. Also, late-stage functionalization of drug relevant molecules, gram scale reaction and application in the treatment of breast cancer cells of the germanium-modified ospemifene analogue highlight the synthetic potency of this reaction. In addition, a realistic mechanism proposal is given in Figure 6d, which seems plausible. Given the importance of organogermanium compounds and the novelty of reductive gemylative alkylation of alkenes, this protocol is an important contribution in organogermane chemistry. Therefore, I'm happy to recommend the publication of manuscript in Nat. Commun., after some revisions were addressed.

1. In the Figure 2, a broad scope of alkyl bromides was shown, how about other alkyl halides, such as alkyl chlorides and alkyl iodides?
2. In this manuscript, the authors demonstrated activated olefins having electron-withdrawing groups, such as acrylates could be tolerated well. What happens with other alkenes, such as styrenes. Getting other things to work would greatly strengthen the study.
3. Ref. 8 and ref. 25 were the same.
4. In the Figures 2-4, neither figure contains "a" labelling, but both captions contain text for "a".
5. In the SI, the "DMA" and "TDAE" were used in section 2 directly, which should be provided their full names to avoid confusing understanding.
6. compounds 1bq and 1bs were missing of HRMS, if they are unknown compounds.
7. In the section 3.2.1 General procedure of SI, it should provide a general equation, not a specialized one. Furthermore, the loading amount of alkene, germanium reagent and Mn powder in the main text should be modified, as well as the ligand should be deleted.

Reviewer #2:

Remarks to the Author:

In this manuscript Zhang and co-worker reported a synthetic protocol for alkylgermane via a Ni-catalyzed atom transfer radical addition and reductive coupling route.

Regarding the chemistry described here, current research is an extension of a well-established area of the chemistry of Ni-catalyzed ATRA and/or reductive coupling. A very similar example of such reaction route has been reported by the authors in this journal. The difference of the two reactions is the use of electrophile, chlorosilane for the previous one and chlorogermane for the current one. Note that both chlorosilane and chlorogermane are of very similar reactivity (oxidative addition) toward Ni(I), which have been reported by several groups in recent years. Furthermore, prior to the authors' works, examples for use of carbo-electrophiles (e.g. alkyl or aryl halides) in such Ni-catalyzed tandem transformations were also reported.

The major merit of this work is the synthetic efficiency and diversity for some organogermanes, especially for preparation of alkyl germane. Addition of germyl radical or anion species (e.g. germymetal reagent) to electron-deficient alkenes usually led to beta-germylation. In this work, alpha-germylation is accomplished, which is complementary to known alkene germylation protocols. On the other hand, peralkylgermyl normally is difficult to be installed via an anion precursor since peralkylgermylmetal reagents are hard to be prepared so far. Recent reports show that

peralkylgermane is a robust and easy-handling precursor for either carbo-anion or radical in some potentially useful synthetic transformations. Indeed, synthetic organogermanium chemistry is rather less developed compared to those of organosilicon and -tin. In this regard, current work is useful for syntheses of some alkylgermane derivatives and hence would be valuable for future research on organogermanium reactions.

In general, this work might be of interest in the areas of organogermanium chemistry, organogermanium-mediated molecular transformations, and/or germanium-centered functional/pharmaceutical molecules and hence might be published in this journal.

Technically, current work is solid, with an extensive examination of reaction scope and careful mechanistic study. However, the authors should also provide enough details for negative results, such as unreactive alkyl bromide and alkenes. For example, only one example was described for alpha-substituted acrylate (methyl). What happens with other substituents? Furthermore, what happens with beta-substituted acrylate (cis- or trans-, cyclic or acyclic, etc.)? What happens with other alkyl halides (e.g. iodide)?

Reviewer #3:

Remarks to the Author:

The paper by Prof Zhang, Bao and colleagues reports the synthesis of alkylgermanes via a nickel-catalysed germylative alkylation of electron-deficient olefins. The reaction enables the formation of a broad range of alkylgermanes from primary, secondary, or tertiary alkyl bromides, different activated alkenes (i.e., bearing ester, amide, cyano, sulfone, phosphonate moieties) and chlorotrialkylgermanes, in the presence of nickel bromide and manganese. The scope of the reaction seems to be broad, encompassing a wide range of substituents and useful functional groups. Examples of functionalisation of the synthesised alkylgermanes have been reported. A number of control experiments to investigate the reaction mechanism have been performed and a reasonable hypothesis of mechanism is provided. The potential synthetic utility of the described protocol is studied by its application to the late-stage functionalisation of pharmacologically relevant molecules, including ospemifene, which demonstrated interesting bioactivity in the prevention of breast cancer, behaving as Selective Estrogen Receptor Modulator (SERM).

In my opinion, dealing with the development of a new route towards functionalised and functionalisable germanium-containing molecules, the results described in the study could meet the interest of a broad range of organic chemists. The versatility of organogermanes in organic synthesis and their potential application as bioisosteres of carbon-analogues in medicinal chemistry render this work attractive for a broad readership. On the basis of these considerations, I feel that this paper can be suitable for publication in Nat. Commun. after revision. In my opinion, there are several points that should be addressed before accepting the manuscript for publication (see below).

About the synthetic part:

- During the optimisation of the reaction conditions, the authors reported interesting yields using the chiral bisoxazoline ligand (L5); see for example entry 6 of Table 1. However, this reviewer could not find any data on the stereochemical outcome of the reaction in the presence of the chiral ligand. Is compound 4 formed as a racemate? Enantioselective HPLC (or SFC) should be performed on the alkylgermane 4. The possibility to develop an enantioselective approach towards alkylgermanes would significantly enhance the impact of the work.
- Lines 180-181: "to construct germanium-containing all-carbon quaternary stereogenic center 40-45...". Please, consider the structure of the mentioned and revise the sentence. "all-carbon quaternary stereogenic center" is not correct.
- In the supplementary information file, the scale of all provided NMR spectra should include the 'zero'

(zero ppm).

About the biological part:

The authors reported a study on the antiproliferative activity of ospemifene and a germanium-containing related compound (71) against MCF-7 (ER+) and MDA-MB-231 (ER-) human breast cancer cell lines using MTT assay. The metabolic stability of 71 was also explored and compared with that of ospemifene by mouse liver microsomal metabolic assay. The hemocompatibility of the germanyl-derivative of ospemifene was also assessed. Globally, the obtained results are encouraging and, although 71 exhibited only a moderate improvement in IC50 values with respect to ospemifene against MCF-7 (ER+) cell line, the functionalisation of the ospemifene structure seems to be related with improved pharmacological properties. However, although the preliminary findings reported are promising, additional studies should be performed to highlight some important points.

- The toxicity of compound 71 against normal human cell lines (breast, liver, etc.) should be evaluated and compared with that of ospemifene.

- Although ospemifene is an achiral molecule, compound 71 possesses a stereogenic center. The activity of the ospemifene-derivative 71 has been investigated on the racemate; however, the biological properties of the two enantiomers of 71 can be significantly different. This point needs to be addressed by assessing the biological properties of the two enantiomers of 71. Separation of racemate 71 should be accomplished (for example by enantioselective HPLC) and the antiproliferative properties of the two enantiomers against MCF-7 (ER+) and MDA-MB-231 (ER-) cells – as well as cytotoxicity studies against normal cells – should be investigated.

- The structure of 71 differs from ospemifene not only for the presence of the germanyl substituent. Is there any data on the antiproliferative data of the compound that could be prepared through protodegermanation of 71?

Please, avoid using terms such as "great yields". Stereochemistry indicators (R, S) should be italicised (i.e., line 207).

The English form should be revised as there are some typing/grammar mistakes throughout the text.

Point-by-point responses to the comments

Reviewer: 1

Recommendation: ‘Therefore, I’m happy to recommend the publication of manuscript in Nat. Commun., after some revisions were addressed.’

Comments:

1. In the Figure 2, a broad scope of alkyl bromides was shown, how about other alkyl halides, such as alkyl chlorides and alkyl iodides?

Response: Thanks for your comments. We selected 1-chloro-3-phenylpropane and 1-iodide-3-phenylpropane as the substrates for the germylative alkylation with 2a and 3a under standard conditions, which gave desired products in 0% and 52% yields, respectively. These results indicate that alkyl chlorides are inert, but alkyl iodides are reactive in this protocol. The related data has been added in the revised manuscript and the supplementary information.

2. In this manuscript, the authors demonstrated activated olefins having electron-withdrawing groups, such as acrylates could be tolerated well. What happens with other alkenes, such as styrenes. Getting other things to work would greatly strengthen the study.

Response: Thanks a lot for your kind suggestions. Several different types of olefins including styrene, α -substituted acrylate, β -substituted acrylates and unactivated terminal alkene were tried, and the results demonstrated that only α -substituted acrylate is compatible in current system, which gives the desired product in 67% yield (entry 4 of following table). The related data has been added in the revised manuscript and the supplementary information.

Entry	T.M. (%)	alkyl halide (%)	4b	4c	4d	Hydroalkylation	4f
1	/	/	14	12	/	/	4
2	/	/	35	36	/	6	3
3	/	/	31	8	1	10	1
4	67 (isolated)	/	13	6	5	/	1
5	/	/	43	20	4	5	4
6	/	/	14	32	/	12	4
7	/	/	39	3	/	/	14

3. Ref. 8 and ref. 25 were the same.

Response: Sorry for this mistake. The ref. 25 has been changed to another reference ‘Pal, P. P., Ghosh, S. & Hajra, A. Recent advances in carbosilylation of alkenes and alkynes. *Org. Biomol. Chem.* 21, 2272–2294 (2023)’ in the revised manuscript.

4. In the Figures 2-4, neither figure contains “a” labelling, but both captions contain text for “a”.

Response: Thanks for pointing these out. We have made the appropriate modifications in the revised manuscript.

5. In the SI, the “DMA” and “TDAE” were used in section 2 directly, which should be provided their full names to avoid confusing understanding.

Response: Thanks for your comments. The full names of ‘DMA’ as ‘*N,N*-dimethylacetamide’ and ‘TDAE’ as ‘tetrakis(dimethylamino)ethylene’ have been added in the revised supplementary information.

6. compounds 1bq and 1bs were missing of HRMS, if they are unknown compounds.

Response: Thanks for pointing out the missing data. The HRMS data of 1bq and 1bs have been added in the revised supplementary information.

7. In the section 3.2.1 General procedure of SI, it should provide a general equation, not a specialized one. Furthermore, the loading amount of alkene, germanium reagent and Mn powder in the main text should be modified, as well as the ligand should be deleted.

Response: Thanks for your kind reminder. In the revised supplementary information, the general equation has been provided, the loading amount of alkene, germanium reagent and Mn powder in the main text have also been modified, as well as the ligand has been deleted.

Reviewer: 2

Recommendation: In general, this work might be of interest in the areas of organogermanium chemistry, organogermanium-mediated molecular transformations, and/or germanium-centered functional/pharmaceutical molecules and hence might be published in this journal.

Comments:

1. However, the authors should also provide enough details for negative results, such as unreactive alkyl bromide and alkenes. For example, only one example was described for alpha-substituted acrylate (methyl). What happen with other substituents? Furthermore, what happens with beta-substituted acrylate (cis- or trans-, cyclic or acyclic, etc.)?

Response: Thanks a lot for your good comments. Based on our further attempts, the unsuccessful examples of alkyl bromides were shown as following. Functional groups, such as hydroxyls, aldehydes, alkynes, triazoles and diaryl ketones are not compatible in this protocol. Furthermore, amide group can't be tolerated well in this reaction, only less than 10% of target product was detected by crude NMR with the assistance of additional ligand 5. In most cases, dehalogenation and dehydrohalogenation of alkyl bromides were detected as the byproducts. The related results have been added in the revised manuscript and the supplementary information.

For the olefins, dimethyl itaconate also showed good reactivity as an another α -substituted acrylate, and gave the desired product in 67% isolated yield. However, the β -substituted acrylates including methyl crotonate, 5,6-dihydro-2H-pyran-2-one, methyl 1-cyclohexene-1-carboxylate and methyl 1-cyclopentene-1-carboxylate, were not compatible under standard conditions, which gave no target products along with byproducts from dehalogenation, dehydrohalogenation, cross-coupling, hydroalkylation and homocoupling of starting materials detected by GC-FID and NMR. The related results have been presented in the revised manuscript and the supplementary information.

3. What happen with other alkyl halides (e.g. iodide)?

Response: Thanks for your nice suggestions. We selected 1-chloro-3-phenylpropane and 1-iodide-3-phenylpropane as the substrates for the germylative alkylation with 2a and 3a under standard conditions, which gave desired products in 0% and 52% yields, respectively. These results indicate that

alkyl chlorides are inert, but alkyl iodides are reactive in this protocol. The related data has been added in the revised manuscript and the supplementary information.

Reviewer: 3

Recommendation: On the basis of these consideration, I feel that this paper can be suitable for publication in Nat. Commun. after revision. In my opinion, there are several points that should be addressed before accepting the manuscript for publication.

Comments:

1. During the optimisation of the reaction conditions, the authors reported interesting yields using the chiral bisoxazoline ligand (L5); see for example entry 6 of Table 1. However, this reviewer could not find any data on the stereochemical outcome of the reaction in the presence of the chiral ligand. Is compound 4 formed as a racemate? Enantioselective HPLC (or SFC) should be performed on the alkylgermane 4. The possibility to develop an enantioselective approach towards alkylgermanes would significantly enhance the impact of the work.

Response: Thanks a lot for the reviewer's nice comments. We firstly failed to get the good separation of the enantiomers for product 4 on enantioselective HPLC. Gratifyingly, the two isomers could be separated after compound 4 reduced to corresponding alcohol under following conditions (Column: OD-H; Mobile Phase: Hexane:iPrOH= 93:7; Flow rate: 1 mL/min; Measure Wavelength: 254 nm). The HPLC result demonstrates that there is no stereoselectivity utilizing the

chiral bisoxazoline ligand (L5). Actually, the investigation on the asymmetric version of this reaction is ongoing in our laboratory.

2. Lines 180-181: “to construct germanium-containing all-carbon quaternary stereogenic center 40-45...”. Please, consider the structure of the mentioned and revise the sentence. “all-carbon quaternary stereogenic center” is not correct.

Response: Thanks a lot for pointing this mistake out. ‘to construct germanium-containing all-carbon quaternary stereogenic center 40-45’ has been corrected into ‘to construct quaternary carbon center-containing alkyl germanium compounds 40-45’ in the revised manuscript.

3. In the supplementary information file, the scale of all provided NMR spectra should include the ‘zero’ (zero ppm).

Response: Thanks for your comments. We have updated all the NMR spectra containing zero ppm in the revised supplementary information.

4. The toxicity of compound 71 against normal human cell lines (breast, liver, etc.) should be evaluated and compared with that of ospemifene.

Response: Thanks for the reviewer’s good suggestions. MCF-10A were chosen as the normal human cell line to evaluate and compare the toxicity between ospemifene and ospemifene-Ge-OH. As shown in the following figure, the semi-lethal dose of ospemifene and ospemifene-Ge-OH were 53.25 μ M and 158.36 μ M independently, which indicated that drugs modified by germanium group have

less toxic side effects. The related results have been provided in the revised manuscript and the supplementary information.

5. Although ospemifene is an achiral molecule, compound 71 possesses a stereogenic center. The activity of the ospemifene-derivative 71 has been investigated on the racemate; however, the biological properties of the two enantiomers of 71 can be significantly different. This point needs to be addressed by assessing the biological properties of the two enantiomers of 71. Separation of racemate 71 should be accomplished (for example by enantioselective HPLC) and the antiproliferative properties of the two enantiomers against MCF-7 (ER+) and MDA-MB-231 (ER-) cells – as well as cytotoxicity studies against normal cells – should be investigated.

Response: Thanks very much for your great comments, we totally agree with that. We do try our best to separate the two enantiomers of ospemifene-Ge-OH by our own laboratory and even by DAICEL CHIRAL TECHNOLOGIES (CHINA) CO., LTD (all the chiral columns they own have been tried), but unfortunately, all attempts failed. We are very sorry for the further cytotoxicity studies couldn't be carried out at current stage. The results in hand really provided a positive signal that the introduction of germanium groups could enhance the antiproliferative properties of the drug, which is recognized to support the conclusion in this article. We appreciate again for these meaningful and enlightening comments, which will be important guidelines for our future research.

6. The structure of 71 differs from ospemifene not only for the presence of the germanyl substituent. Is there any data on the antiproliferative data of the compound that could be prepared through protodegermanation of 71?

Response: Thanks a lot for this good comments. Degermylative compound ospemifene-OH was prepared by the following pathway, and its antiproliferative data against MCF-7 and MDA-MB-231 cell lines were 94.09 μ M and 98.70 μ M respectively. These results clearly demonstrate that the improved anti-cancer activity is resulted from the introduction of germanium group. The related results have been provided in the revised manuscript and the supplementary information.

7. Please, avoid using terms such as “great yields”. Stereochemistry indicators (R, S) should be italicised (i.e., line 207).

Response: Thanks for your kind reminder. We have changed “great yields” to “good yields”. Furthermore, stereochemistry indicators (R, S) have also been italicised in the revised manuscript.

8. The English form should be revised as there are some typing/grammar mistakes throughout the text.

Response: Thanks for your suggestion. We feel sorry for our mistakes, however, we do invite a native English speaker (Prof. Andrew McNally from Colorado

State University) to help polish our draft. And we hope the revised manuscript could be acceptable for you.

Reviewers' Comments:

Reviewer #1:

Remarks to the Author:

The authors have addressed the problems I raised. I believed the present manuscript could be published at the present stage.

Reviewer #2:

Remarks to the Author:

The authors have carefully revised the manuscript and have addressed all issues of my last comments. I suggest the publication of this revised manuscript as it is.

Reviewer #3:

Remarks to the Author:

In my opinion, the authors have satisfactorily addressed the concerns raised by me and the other reviewers. I believe that the manuscript can be accepted for publication in Nature Communications in this form.

Point-by-point responses to the comments

Reviewer #1 (Remarks to the Author):

Comments:

The authors have addressed the problems I raised. I believed the present manuscript could be published at the present stage.

Response: Thanks again for the reviewer's good comments and kind affirmation on our work.

Reviewer #2 (Remarks to the Author):

Comments:

The authors have carefully revised the manuscript and have addressed all issues of my last comments. I suggest the publication of this revised manuscript as it is.

Response: Thanks for your affirmation on our research, and it's really a nice encouragement for us.

Reviewer #3 (Remarks to the Author):

Comments:

In my opinion, the authors have satisfactorily addressed the concerns raised by me and the other reviewers. I believe that the manuscript can be accepted for publication in Nature Communications in this form.

Response: We appreciate a lot for this reviewer's previous and current suggestions, they do make our work better.